# Incorporation of lumped IVOC emissions into the ORACLE model (V1.1): A multi-product framework for assessing global SOA formation from internal combustion engines

Susanne M. C. Scholz[1], Vlassis A. Karydis[1], Georgios I. Gkatzelis[1], Hendrik Fuchs[1,4], Spyros N. Pandis[2,3], and Alexandra P. Tsimpidi[1]

[1]Institute of Energy and Climate Research: Troposphere (ICE-3), Forschungszentrum Jülich GmbH, Jülich, 52428, Germany
[2]Department of Chemical Engineering, University of Patras, Patras, Greece
[3]Institute of Chemical Engineering Sciences (ICE-HT), FORTH, Patras, Greece
[4]Department of Physics, University of Cologne, Cologne, Germany

*Correspondence to*: Alexandra P. Tsimpidi (a.tsimpidi@fz-juelich.de)

**Abstract.** Secondary organic aerosol (SOA) is a major component of particulate matter but is often underpredicted in chemistry climate models. Recent advances in measuring and resolving the chemically complex structure of intermediate volatile organic compounds (IVOC) have shown that IVOCs, despite their high SOA yields, have long been underrepresented in models. These compounds are key precursors of SOA from emissions in the road transport sector and significantly influence SOA formation. Understanding vehicle emissions, their chemistry, and their SOA-forming potential is essential for accurately estimating their contributions to atmospheric SOA and global organic aerosol loads. To improve this understanding, we have updated the organic module ORACLE in the global chemistry climate model EMAC. The existing IVOC representation was based on scaled organic carbon (OC) emissions and a highly parameterized volatility basis set (VBS) which underestimated global IVOC emissions, particularly those from gasoline combustion, and oversimplified their chemistry. Here, we replaced this approach with a lumped species framework, in which experimental data for gasoline and diesel emissions were grouped into seven lumped species based on their chemical properties and hydroxylation potentials. These species were linked to adjusted emission inventories for regional diesel and gasoline consumption. A 10-year simulation with the updated ORACLE-IVOC model resulted in significant changes. The global atmospheric burden of road transport IVOC-derived SOA (SOA-iv) increased by 1 order of magnitude, from 0.014 Tg to 0.13 Tg. The composition of road transport organic aerosol (OA) shifted, with SOA-iv contributing 2.5 to 13 times more than the primary organic aerosol (POA) and SOA derived from semi-volatile organic compounds combined. In the results using the previous model, this ratio was between 0.4 and 1.1. The geographical distribution of OA also changed. Regions rich in gasoline relative to diesel emissions experienced higher concentration increases, and remote areas experienced elevated concentrations due to more efficient long-range transport of the new lumped IVOC species. Overall, these changes led to a significant increase in the contribution of road transport to total anthropogenic SOA-iv from an average value of 3% to 35%.

# 1 Introduction

Secondary organic aerosol (SOA) is formed when volatile organic compounds (VOCs) undergo one or more chemical transformations in the gas phase to less volatile compounds, and finally pass into particle phase through nucleation, condensation or multiphase chemistry (Donahue et al., 2006; Srivastava et al., 2022). As one of the major components of particulate matter (PM), SOA contributes significantly to air pollution and the associated health risks (Zhang et al., 2007; Szopa et al., 2021; Pope et al., 1995; WHO, 2021). SOA has also been shown to have the highest association per unit mass with cardiorespiratory mortality (Pye et al., 2021) probably due to its high oxidative potential (Baulig et al., 2003; Li et al., 2003; Pye et al., 2021). In addition, SOA affects the Earth's energy budget by contributing to the absorption and scattering of radiation (direct effect) and by acting as cloud condensation nuclei and altering the properties of clouds (indirect effects) (Boucher et al., 2013). However, SOA, especially from anthropogenic precursors, is the least understood major PM component due to the enormous diversity and complexity of precursors and oxidation products (Szopa et al., 2021; Srivastava et al., 2022). Therefore, accurate modelling and assessment of the magnitude and variability of atmospheric SOA is a major challenge and an essential task for improving our understanding of these important aerosols.

A vital factor to consider in this quest is the chemical structure and volatility of the SOA precursors, as this influences their atmospheric transport and deposition, as well as their oxidation and aerosol formation potential (SOA yields) (Robinson et al., 2007; Srivastava et al., 2022). The SOA precursors can be classified into four groups based on their volatility (Robinson et al., 2007; Donahue et al., 2009; Tsimpidi et al., 2014). The most volatile species are the volatile organic compounds with saturation concentrations ($C^*$, at 298 K and 1 atm) greater than $10^6 \, \mu g \, m^{-3}$. With increasing carbon number, the saturation concentration and the volatility of organic molecules usually decrease. Therefore, species with about 12 or more carbon atoms and $C^*$ approximately between $10^3$ and $10^6 \, \mu g \, m^{-3}$ are classified as intermediate volatility organic compounds (IVOCs) (Robinson et al., 2007; Donahue et al., 2009; Zhang et al., 2022b). In most ambient atmospheric conditions, IVOCs are still almost exclusively (> 99%) in the gas phase. At low temperatures around 0°C and high organic aerosol (OA) loadings of $10 \, \mu g \, m^{-3}$, still about 94% of IVOCs are in the gas phase (Lu et al., 2018). Semivolatile organic compounds (SVOCs) with $C^*$ in the order of 1 to $10^2 \, \mu g \, m^{-3}$ exist in both the particulate and gaseous phases. The low volatility organic compounds (LVOCs) exist almost exclusively in the particulate phase due to their low saturation concentration in the order of $C^* \leqslant 10^{-1} \, \mu g \, m^{-3}$ (Robinson et al., 2007; Donahue et al., 2009; Tsimpidi et al., 2016; Lu et al., 2018). The most abundant of these compounds are the VOCs of anthropogenic (e.g., aromatics, alkanes, olefines, etc.) and biogenic (e.g., isoprene and terpenes) origin (Pandis et al., 1992; Odum et al., 1996; Kanakidou et al., 2005; Srivastava et al., 2022). Although IVOCs have considerably lower emission loads than VOCs, they undergo oxidation more efficiently and have higher SOA yields due to their lower volatility (Zhao et al., 2014; Docherty et al., 2021; Zhang et al., 2022a). This makes IVOCs equally or even more important SOA precursors, particularly in urban environments, as evidenced by ambient monitoring, smog chamber experiments (Sartelet et al., 2018; Jorga et al., 2019), computational chemical mechanism studies (Lu et al., 2018; Pye et al., 2023), and modelling efforts that explicitly account for IVOC emissions from combustion sources (Jathar et al., 2014). IVOCs thus represent a chemically

diverse and relatively under-characterized class of SOA precursors, whose role in atmospheric aerosol formation remains
uncertain.

Recent studies of IVOCs have revealed a variety of important sources, ranging from wildfires and residential biomass burning (Schauer et al., 2001; Tang et al., 2022; Huang et al., 2022) to waste management processes (Fujitani et al., 2020 and 2023), cooking (Song et al., 2022), oil sands mining and oil leaks (Liggio et al., 2016; Sommers et al., 2022 and De Gouw et al., 2011), chemical (consumer) products (Li et al., 2018; Seltzer et al., 2021),  liquid or even solid fuel combustion (Cai et al.,
2019; Tang et al., 2022; Qian et al., 2021, respectively). Among these, ship emissions (Kangasniemi et al., 2023; Huang et al., 2018), and in particular, road transport stands out as some of the most extensively studied IVOC sources. Globally, the road transportation sector accounts for approximately 9% of total IVOC emissions (Huang et al., 2023). However, in major urban centers such as Beijing and London, road transport has been reported to contribute a significant fraction of total IVOCs, with contributions varying depending on fleet composition and fuel type (Schauer et al., 1999; Drozd et al., 2021; Zhang et al.,
2022a; Bessagnet et al., 2022). For instance, diesel-related IVOCs from both road and non-road engines accounted for 47% of total IVOC emissions in Guangzhou, a coastal city in China (Fang et al., 2022), and were responsible for 30% of total SOA in London in 2012 (Ots et al., 2016).

For modelling the formation and evolution of POA and SOA in the atmosphere, sector specific emission inventories of organic carbon (OC) and VOCs are generally available. However, commonly used measurement techniques have limitations
in capturing intermediate volatility organic compounds (IVOCs). Filter-based methods typically detect only particulate OC species with saturation concentrations up to $10^4$ $\mu g$ $m^{-3}$, while gas chromatography-based techniques primarily resolve lighter VOC species with $C^* > 10^6$. Neither method is well-suited to fully detect and characterize the volatile yet chemically complex IVOCs (Shrivastava et al., 2008; Robinson et al., 2007, 2010; Tsimpidi et al., 2014; Zhao et al., 2014), which may include hundreds of isomers (Fraser et al. 1997, Goldstein et al., 2007; Zhao et al., 2014). Consequently, there is an information gap
in the inventories for organic compounds with a saturation concentration of about $10^4 < C^* \leqslant 10^6$ $\mu g$ $m^{-3}$ (Tsimpidi et al., 2014; Lu et al., 2018) and most IVOCs from anthropogenic combustion sources are missing from common inventories (Srivastava et al., 2022). To account for the missing IVOCs, many chemical transport models estimate IVOC emissions assuming that they are proportional to the OC emissions using enhancement factors of typically 1.5, but up to 6.5 in combination with the volatility basis set (VBS) framework (Robinson et al., 2007; Shrivastava et al., 2008; 2011; Hodzic et
al., 2010; Jathar et al., 2011; Tsimpidi et al., 2010, 2011, 2014, 2016, 2017). The VBS is commonly used by chemistry climate models to simulate the chemical evolution of IVOCs by distributing IVOC emissions over logarithmically spaced volatility bins and participating in chemical reactions with the hydroxyl radical, OH (Donahue et al., 2006; Murphy and Pandis, 2009; Tsimpidi et al., 2010). The products of this oxidation are 1 or more orders of magnitude lower in saturation concentration and can more easily condense to the aerosol phase to form SOA (Shrivastava et al., 2008; Tsimpidi et al., 2017).
Estimating a fixed scaling factor for IVOC emissions based on the emitted OC is not a universally valid, realistic approach as the scaling factor varies significantly depending on the emission source type and since the correlation of IVOC to OC is weaker than to VOC (Zhao et al., 2015, 2016; Lu et al., 2018). Furthermore, the accurate description of SOA formation and

composition by simply knowing the volatility or saturation concentration classification is not sufficient, as the number of carbon atoms, the molecular weight and especially the chemical structure also strongly influence the chemical reaction rates

and thus the final yield (Gentner et al., 2012; Manavi and Pandis, 2022; Srivastava et al., 2022). For example, a cyclic structure is responsible for less volatile ring-opening products, which, together with a tendency to form oligomers, increases SOA yields. Therefore, branched alkanes have lower SOA yields than linear alkanes, which in turn have lower yields than cyclic alkanes (Srivastava et al., 2022). This means that branched or cyclic hydrocarbons react at different rates despite having the same carbon number (Zhao et al., 2014). In the case of aromatics, the structure also influences the SOA yield, which depends more

on the position of the substituent than on the length of the compound (Srivastava et al., 2022). Therefore, the fact that each VBS surrogate species must represent thousands of individual, differently structured IVOCs, for which the same reaction rate and oxidation pathway are then assumed, is an oversimplification and a weakness in modelling SOA from IVOC-rich sources.

Recently, new approaches have been applied to speciate the missing IVOCs. An important example is IVOCs from internal combustion engines, which have been increasingly studied by several laboratory, field and computational studies. Gentner et

al. (2012) and Zhao et al. (2014, 2015, 2016) have characterized the emissions of vehicles representative for the car fleet in California, Liu et al. (2021) and Zhang et al. (2022b) for China, Marques et al. (2022) and Sarica et al. (2023) for the one in Europe. These studies usually report their emission factors of IVOCs as a fraction of VOCs due to their good correlation. The study of 79 emission factors of speciated and unspeciated IVOCs from both diesel and gasoline combustion by Zhao et al. (2015, 2016) has served as a database for several model approaches that have taken up the challenge of more accurately

describing the complex chemical evolution of IVOCs in the atmosphere. For example, Sartelet et al. (2018) used this dataset to estimate IVOC and SVOC traffic emissions over Paris. Compared to traditional IVOC/POA emission factors, their approach led to an 8% decrease in simulated OA concentrations along motorways, but a 25% increase in OA concentrations across the greater Paris urban area. As a second example, Lu et al. (2020) categorized 79 IVOC species, whose emission factors were measured by Zhao et al. (2015, 2016), based on their aromatic or aliphatic structure and further subdivided them by volatility,

resulting in four lumped alkane species and two lumped aromatic species. This framework was used to model SOA concentrations over California and Nevada during May and June 2010, leading to an approximate 70% increase in the daily SOA peak. This approach was subsequently refined by Manavi and Pandis (2022), who incorporated more detailed chemical structure information, including carbon number, branching, aromatic complexity (e.g., polycyclic vs. monocyclic), and OH reactivity. Their updated scheme resulted in a slightly different grouping of the four alkane lumped species and a more specific

classification of aromatic IVOCs into two lumped species of polycyclic aromatic hydrocarbons and one lumped species of simple aromatics with $C_{11}$–$C_{22}$ benzenes. Although volatility was not explicitly used as a lumping criterion in their framework, it was implicitly accounted for through molecular weight and structural characteristics. Using this refined lumping approach, they simulated SOA formation over Europe during early summer.

The enhanced chemical specificity in the Manavi and Pandis (2022) framework allows for a more accurate representation

of reaction rates and SOA yields, both of which are strongly influenced by molecular structure (Gentner et al., 2012; Srivastava et al., 2022). In this work, we have adopted this framework and implemented it into ORACLE, a computationally efficient

module for simulating the composition and evolution of organic aerosol in the atmosphere (Tsimpidi et al., 2014). This integration allows for a more realistic and comprehensive consideration of IVOCs from the transport sector, viewed from a global and long-term perspective. The implementation and the resulting SOA simulations, compared to those produced using the previously employed VBS framework, are presented and discussed in this study.

## 2 Methods

In this study, the computationally efficient submodel ORACLE (Tsimpidi et al., 2014), which simulates the OA composition and evolution in the atmosphere, is further developed to incorporate a new IVOC scheme for simulating the SOA formation from on-road transport emissions. Specifically, a lumped species approach for IVOC emissions from the road transport sector is implemented into the base version of ORACLE (hereafter referred to as ORACLE-base), resulting in the updated model version ORACLE-IVOC.

Both ORACLE-base and ORACLE-IVOC are coupled with the ECHAM5/MESSy Atmospheric Chemistry (EMAC) model (Jöckel et al., 2005) to simulate the period 2010 – 2020, using 2010 as a spin-up year. The simulations are conducted at a horizontal resolution of approximately 1.86°×1.88° (latitude × longitude) with 31 vertical layers extending up to 10 hPa (~25 km above the surface), corresponding to the T63L31 configuration. Anthropogenic emissions from all sectors are derived from the CAMS global anthropogenic emissions inventory (CAMS-GLOB-ANT; Soulie et al., 2023). In both ORACLE-base and ORACLE-IVOC configurations, the formation of SOA from VOC, IVOC, and SVOC precursors is tracked separately. Additionally, emissions from the road transport sector are treated independently from other sectors for the simulation of IVOC, SVOC, and LVOC, allowing for a targeted analysis of SOA and POA contributions specifically attributable to traffic-related emissions. VOCs from road transport are included as part of the total anthropogenic fossil fuel and biofuel emissions in the CAMS inventory and follow the implementation described by Tsimpidi et al. (2014). ORACLE simulates the photochemical oxidation of seven lumped VOC species considered SOA precursors, including alkenes, aromatics, isoprene, olefins, and monoterpenes, with oxidants such as OH, $O_3$, O, and $NO_3$. The resulting oxidation products are grouped into two categories: aSOA-v (from anthropogenic VOCs) and bSOA-v (from biogenic VOCs). These products are distributed into volatility bins with saturation concentrations of $10^0$, $10^1$, $10^2$, and $10^3$ µg m$^{-3}$ at 298 K, using aerosol mass yields from Tsimpidi et al. (2014). Since VOC-derived SOA is treated identically in both model versions, it does not influence the comparison between them. Including total SOA in the comparison of ORACLE-IVOC and ORACLE-base would obscure the differences specifically attributable to the updated IVOC emissions and chemistry, which are the focus of this study. Therefore, we do not discuss total SOA formation in detail but instead focus on POA and SOA formation from LVOCs, SVOCs, and especially IVOCs from road transport emissions. The POA and SOA formation mechanisms for these three precursors, as implemented in the two model versions, are described in detail in Sections 2.1 and 2.2.

## 2.1 Description of the ORACLE-base module

ORACLE (Tsimpidi et al., 2014) treats both POA and SOA as volatile and reactive and therefore can describe all organics in a unified volatility basis set (VBS) framework (Donahue et al., 2006). Primary emissions include VOCs, IVOCs, SVOCs, and LVOCs, which are attributed to several surrogate species with logarithmically spaced bins of their saturation concentration $C^*$. Except LVOCs, which do not participate in further photochemical reactions due to their low volatility and dominant existence in the particulate phase, all primary organic gases (POGs) can undergo ageing reactions with OH in the gas-phase to form oxygenated secondary organic gases (SOGs), which are then assigned to lower volatility bins in the VBS framework. SOGs and POGs may also partition between the gas and particulate phases to form SOA and POA, respectively, by assuming bulk equilibrium between the two phases and that all organic compounds form a pseudo-ideal solution. Phase partitioning depends on atmospheric conditions such as ambient temperature, the species' enthalpy of vaporization, the pre-existing OA mass, and size distribution, as detailed in Tsimpidi et al. (2014).

For the description of OA from the road transport sector, OC emissions are taken from the CAMS global anthropogenic emissions inventory (CAMS-GLOB-ANT; Soulie et al., 2023) and distributed across six newly introduced POG/POA surrogate species of different volatilities: 9% of OC is attributed to one surrogate species for LVOCs, 23%, 48% and 20% to three SVOC surrogate species, and 50% and 100% to two IVOC surrogate species. The additional 150% of OC emissions accounts for IVOCs that are typically underrepresented in inventories. This volatility distribution is derived from laboratory experiments on diesel exhaust conducted by Robinson et al. (2007) and were fitted to the volatility basis set (VBS) framework as described in Tsimpidi et al. (2016). The distribution, along with the associated volatility bins and ageing reactions, is illustrated schematically in Fig. 1.

OC emissions from other sectors are also based on the CAMS-GLOB-ANT inventory. Similar to the organics from the road transport sector, their evolution is described with the VBS framework (Tsimpidi et al., 2014). The photooxidations and partitioning for IVOCs are described by the following reactions:

$$IVOC_i \ (g) \leftrightarrow POAiv_i \ (p) \qquad \text{(R1)}$$
$$IVOC_i \ (g) + OH \rightarrow 1.15 \ SOGiv_{i-2} \ (g, fresh) \qquad \text{(R2)}$$
$$SOGiv_i \ (g) + OH \rightarrow 1.15 \ SOGiv_{i-2} \ (g, aged) \qquad \text{(R3)}$$
$$SOGiv_i \ (g) \leftrightarrow SOAiv_i \ (p) \qquad \text{(R4)}$$

The index $i$ indicates the saturation concentration with $C^* = 10^i$ µg m$^{-3}$ for each surrogate species. The subsequent oxidation products have a $C^* = 10^{i-2}$ µg m$^{-3}$, i.e., their volatility is reduced by a factor of 100. Chemically, ORACLE is based on the assumption that the OH reactions of IVOCs have a rate constant of $2\times10^{-11}$ cm$^3$ molecule$^{-1}$ s$^{-1}$ (Pye and Seinfeld, 2010) and that two oxygen atoms are added in one oxidation step, leading to the 15% mass increase if an average precursor is assumed to be a C$_{15}$ molecule (Tsimpidi et al., 2014; Atkinson and Arey, 2003). The same equations apply to SVOC$_i$ that can either directly partition into the aerosol phase to become POA-sv$_i$ or oxidise to SOG-sv$_{i-2}$ with SOA-sv$_{i-2}$ being the corresponding

aerosol. To track the contribution of the road transport sector to SOA formation, five new SOG/SOA-iv species and five new
SOG/SOA-sv species were added to the ORACLE-base model.

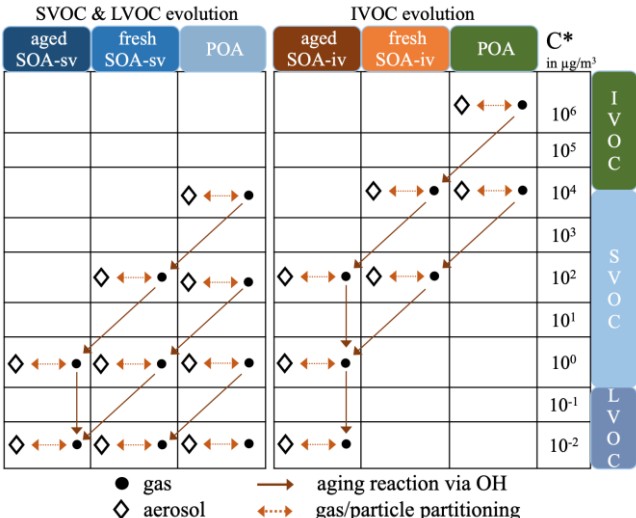

**Figure 1:** Schematic of VBS resolution and POA and SOA formation from SVOCs, LVOCs and IVOCs road transport emissions for
ORACLE-base.

## 2.2 Description of the ORACLE-IVOC module

A key aspect of the method used in this work to improve representation of the IVOC chemistry is the grouping of speciated
and unspeciated IVOCs based on their molecular weight, OH reaction rate constants, and chemical structure. This approach
follows the methodology developed by Manavi and Pandis (2022), which has been largely adopted by ORACLE-IVOC. They
classified 79 IVOC species, identified in emissions from a diverse fleet of gasoline and diesel vehicles during chassis
dynamometer testing (Zhao et al., 2015; 2016), into seven lumped species. These datasets include emission factors of 57
speciated IVOCs, 11 unspeciated $C_{12}$ to $C_{22}$ branched alkanes (b-alkanes), and 11 unspeciated $C_{12}$ to $C_{22}$ cyclic compounds.

This resolution of the IVOC spectrum could be reached by the application of Tenax adsorbent tubes, which can collect
over 90% and 97% of IVOCs from gasoline and diesel exhaust, respectively. For the determination of common VOC emission
studies those tubes are usually not used because they only collect about 5% and 55% of the total organic emissions from
gasoline and diesel exhaust, respectively (Zhao et al., 2015, 2016). After detection, the adsorbent tubes were further analysed
by gas chromatography and mass spectrometry. Finally, the 57 speciated IVOC compounds were found for both diesel and
gasoline, but most of the mass (> 90%) appeared as unresolved complex mixture (UCM) in the chromatograph. For the best
possible characterization, Zhao et al. (2014, 2015, 2016) quantified them into 11 bins depending on their carbon number,
according to the gas chromatography retention time. Based on the mass spectra, the IVOC UCM bins could be subdivided
further into unspeciated branched alkanes and then unspeciated cyclic compounds, which resulted in the distinction of a total

of 79 different IVOC species for both fuel types, as described above. Further details about the choice of this dataset are explained in section 2.3.2.

In diesel emissions, the cyclic compounds are more likely to be cyclic alkanes, whereas in gasoline emissions, they are more likely cyclic aromatics. To reflect this, Manavi and Pandis (2022) assigned the five unspeciated $C_{12}$ to $C_{16}$ cyclic compounds from diesel measurements to alkane lumped species, while those from gasoline measurements were attributed to polycyclic aromatic hydrocarbons (PAH) lumped species. Careful consideration of unspeciated compounds is crucial because their emission factors are significantly higher than those of speciated alkanes and PAH IVOCs. Accurately representing these compounds improves the understanding of IVOC chemistry and their role in SOA formation.

The four lumped species, designated as ALK6, ALK7, ALK8, and ALK9, represent $C_{12}$ to $C_{22}$ alkanes. They include speciated alkanes, unspeciated branched alkanes, and unspeciated cyclic alkanes. Each lumped species corresponds to molecules with specific carbon numbers and reaction rates. For example, ALK6 includes $C_{12}$ to $C_{14}$ alkanes that have reaction rates of $1.3 - 1.8 \ 10^{-11} \ cm^3 \ molecule^{-1} \ s^{-1}$. This group contains both speciated compounds, such as dodecane, tridecane, and tetradecane, as well as unspeciated ones. Alkanes are in fact the major class of anthropogenic VOCs in urban areas and are mostly emitted from combustion sources (Srivastava et al., 2022). The lumped species for aromatics (ARO3) includes only speciated IVOCs, ranging from $C_{11}$ (pentylbenzene) to $C_{22}$ (pentadecylbenzene). The numbering of the alkane (ALK6-9) and aromatic (ARO3) lumped species continues from the VOC chemistry introduced in the earlier version of ORACLE (Tsimpidi et al., 2014), which includes ALK1–5 and ARO1-2. The two PAH lumped species, PAH1 and PAH2, represent $C_{10} - C_{17}$ polycyclic aromatic hydrocarbons. Although many PAHs have saturation concentrations above the typical IVOC range (i.e., $>10^6 \ \mu g \ m^{-3}$), their SOA yields are comparable to other IVOC species (see Table 1), justifying their inclusion as IVOC-like precursors. These seven IVOC lumped species are added to the ORACLE-IVOC as the exclusive IVOC contributors from the road transport sector. The complete mapping of speciated and unspeciated IVOCs to lumped species, along with a detailed explanation of the lumping methodology, is provided in Manavi and Pandis (2022).

**Table 1:** Molecular weight (MW), reaction rate and molecular aerosol mass-based yields for each IVOC lumped species.

| Lumped IVOC | MW (g mol$^{-1}$) | $H_{law}$ constant (M atm$^{-1}$) | Reaction rate constant (cm$^3$ molec$^{-1}$ s$^{-1}$) | Molecular aerosol mass-based yields | | | | |
|---|---|---|---|---|---|---|---|---|
| | | | | $\alpha_{C*=10^{-1}}$ | $\alpha_{C*=10^{0}}$ | $\alpha_{C*=10^{1}}$ | $\alpha_{C*=10^{2}}$ | $\alpha_{C*=10^{3}}$ |
| ALK6 | 183 | | | 0.03 | 0.03 | 0.02 | 0.05 | 0.23 |
| ALK7 | 224 | $2.5\times10^{-4}$ | $9.3\times10^{-12}$ | 0.02 | 0.10 | 0.36 | 0.09 | 0.05 |
| ALK8 | 265 | | | 0.08 | 0.03 | 0.66 | 0.16 | 0.00 |
| ALK9 | 302 | | | 0.09 | 0.03 | 0.76 | 0.18 | 0.00 |
| ARO3 | 188 | | | 0.00 | 0.00 | 0.12 | 0.18 | 0.26 |
| PAH1 | 137 | $1.4\times10^{-1}$ | $2.6\times10^{-11}$ | 0.00 | 0.01 | 0.19 | 0.02 | 0.05 |
| PAH2 | 175 | | | 0.00 | 0.00 | 0.28 | 0.05 | 0.11 |

The aerosol mass-based yields are derived from smog chamber experiments, which inherently account for both
functionalization and fragmentation processes. Consequently, they represent the net effect of all chemical transformations
occurring during oxidation, including the formation of lower-volatility products that contribute to SOA and the fragmentation
of molecules into more volatile species that do not partition into the particle phase. Furthermore, the ambient $NO_x$
concentrations must be considered, as $NO_x$ fundamentally influences the reaction pathways and thus the resulting SOA yields.
Manavi and Pandis (2022) have used the fitting algorithm of Stanier et al. (2008) and single experimental studies to estimate
the $NO_x$-dependent yields for the new lumped IVOCs. For this study, which focuses on road transport, high-$NO_x$ SOA yields
were implemented in ORACLE-IVOC for each lumped IVOC species. These yields, covering oxidation products with five
different saturation concentrations (five-product basis set), are shown in table 1 along with average molecular weights and
reaction rates derived from the SAPRC mechanism (Manavi and Pandis, 2022).

Overall, each lumped IVOC species reacts with OH to produce SOGs across five volatility bins with $C^*$ of 0.1, 1, 10, 100
and $10^3$ µg m$^{-3}$, based on the aerosol mass-based SOA yields for each bin ($\alpha_{i,IVOC}$), where $i$ represents the exponent in $C^*=10^i$
(Table 1):

$$\text{IVOC} + \text{OH} \rightarrow \sum_{i=-1}^{3} \alpha_{i,\,IVOC}\, \text{SOG}_{C*=10^i} \qquad \text{(R5)}$$

Possible reactions with $O_3$ or the nitrate radical, $NO_3$, are not considered here for computational reasons and because the
atmospheric chemical lifetime of most fossil fuel VOCs is strongly determined by their reaction with OH (Srivastava et al.,
2022). Each of the five fresh SOG-iv products is a composite of precursors from multiple lumped species. These compounds
may either partition into the particulate phase to form fresh SOA-iv or undergo additional oxidation steps with OH to produce
aged SOGs. This representation leads to a total of nine new SOG/SOA-iv species in the ORACLE-IVOC module, with the
subsequent photooxidation reactions being modelled using the VBS framework. Here, the multigenerational products of fresh
SOG-iv are simulated as in ORACLE-base, except that the reaction rate with OH is equal to $1\times10^{-11}$ cm$^3$ molecule$^{-1}$ s$^{-1}$ and
only one oxygen atom is added per oxidation step, corresponding to a 7.5% mass increase and reducing the volatility by a
factor of 10. The corresponding reactions are:

$$\text{SOGiv}_i \,(g, \text{fresh}) + \text{OH} \rightarrow 1.075\ \text{SOGiv}_{i-1} \,(g, \text{aged}) \qquad \text{(R6)}$$
$$\text{SOGiv}_i \,(g, \text{aged}) + \text{OH} \rightarrow 1.075\ \text{SOGiv}_{i-1} \,(g, \text{aged}) \qquad \text{(R7)}$$
$$\text{SOGiv}_i \,(g) \leftrightarrow \text{SOAiv}_i \,(p) \qquad \text{(R8)}$$

A schematic overview of this reaction scheme is presented in Fig. 2. For emissions and chemical processing of SVOCs and
LVOCs across all sectors, as well as IVOCs from non-road transport sources, ORACLE-IVOC adopts the same modelling
framework as ORACLE-base.

Furthermore, the Henry's law (H) constants assigned to IVOC species, which describe their ability to partition into water and thereby determine their wet removal efficiency, differ between ORACLE-IVOC and ORACLE-base. In ORACLE-base, high H values of $10^5$ M atm$^{-1}$ are assumed for surrogate IVOC species, consistent with heavier, less volatile compounds that are typically hydrophilic. In contrast, the H constants assigned to the lumped IVOC species in ORACLE-IVOC are much lower (H = $2.5 \times 10^{-4}$ M atm$^{-1}$ for alkanes and H = 0.14 M atm$^{-1}$ for aromatics; Table 1), reflecting their relatively hydrophobic nature, similar to other gaseous VOCs (Jimenez et al., 2009; Pye and Seinfeld, 2010; Hodzic et al., 2014).

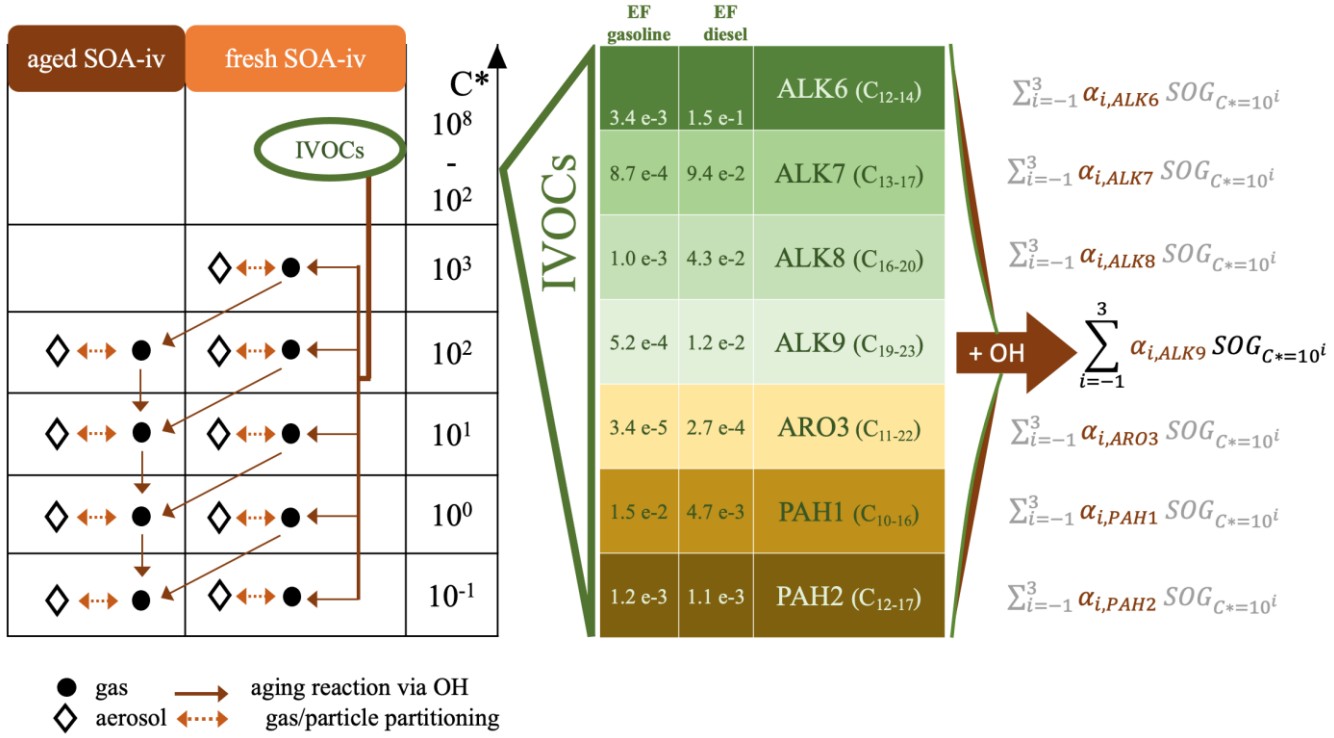

**Figure 2:** Schematic of the IVOC evolution using the lumping scheme. The contribution of each lumped IVOC species from gasoline and diesel is expressed as emission factors (EF) in units of molecules of IVOCs emitted per molecule of VOCs emitted (middle). All lumped species react with OH. The first ageing step to fresh SOG products with different volatilities and saturation concentration $C^*$ at 298 K is mathematically described by the sum equations (right side) with the aerosol mass-based yields $\alpha_i$ for all 7 lumped species. Further ageing to aged SOG and partitioning is described by Reactions R6 – R8. On the left side, the same processes are shown as a vector scheme, with gas and aerosol phases, ageing and partitioning as in Fig. 1.

Overall, ORACLE-IVOC introduces both a revised emission inventory and an updated chemical mechanism. Therefore, the comparison against ORACLE-base presented in this work reflects the combined impact of enhanced emissions and improved chemical treatment. It is not possible to isolate the individual contribution of each component to the overall improvement in SOA predictions, as the previous OC-based scaling approach used in ORACLE-base to estimate IVOC

emissions cannot be directly applied to the new lumped IVOC species due to the lack of necessary chemical speciation. The developed model framework allows for future extensions to include IVOC emissions from other sectors. However, other sectors, such as volatile chemical products (McDonald et al., 2018; Seltzer et al., 2021) and biomass burning (Hatch et al., 2015; Koss et al., 2018), emit IVOCs with distinct chemical structures not typically found in transport emissions (e.g., phenols, furans, cresols, etc.). Therefore, accurately representing IVOC emissions from additional sources will require further research to compile sector-specific emission profiles and classify emitted compounds into appropriate lumped species based on their chemical structure and reactivity.

## 2.3 Estimation of global IVOC emissions from the road transport sector

In ORACLE-IVOC, emissions of IVOCs from the road transport sector are estimated proportionally to the total VOC emissions reported in the global anthropogenic CAMS-GLOB-ANT inventory (Soulie et al., 2023). Since the composition and quantity of the emitted IVOCs vary with the relative share of diesel and gasoline vehicles due to their fuel-dependent emission characteristics, region-specific fleet compositions have been incorporated in the model to more accurately scale the IVOC emissions. This approach ensures a globally consistent, yet regionally differentiated, representation of IVOC emissions from road transport.

### 2.3.1 Global fleet distribution

Organic emissions from the road transport sector originate from the combustion of gasoline and diesel fuels, and multiple non-combustion sources such as tire and brake wear, road asphalt abrasion, and the evaporation of lubricating oil, diesel and gasoline. Most of the non-combustion emissions are typically less volatile and dominate the POA emissions of motor vehicles (Worton et al., 2014). Emissions from road asphalt can include IVOCs under sunny and hot weather conditions, though this source remains poorly quantified and is not currently represented in the CAMS-GLOB-ANT inventory (Sarica et al., 2023; Khare et al., 2020). Furthermore, VOC emissions from the road transport sector are typically approximated as originating solely from the exhaust of gasoline and diesel combustion. This assumption is reasonable on a global scale, where fuel combustion has remained the dominant source of road transport-related VOC emissions over the past decade. Nevertheless, evaporation emissions from diesel (Drozd et al., 2021) and gasoline, particularly during vehicle parking, refueling, and hot soak events (Liu et al., 2015; Sartelet et al., 2023), also represent important sources of VOCs and IVOCs. These evaporation-related emissions are generally not included in standard emission inventories, such as CAMS-GLOB-ANT (Soulie et al., 2024), which do not account for VOCs from fuel evaporation beyond idling. Similarly, the dataset by Zhao et al. (2016), used to calculate IVOC/VOC emission factors (see Section 2.3.2), includes measurements under idling conditions but does not capture evaporation scenarios such as permeation or refueling. Consequently, the IVOC/VOC emission factors derived from Zhao et al. (2016) are consistent with the CAMS inventory and can be appropriately applied.

Furthermore, since the relative contributions of diesel and gasoline vary significantly by region due to differences in fleet composition and fuel usage, regional specificity is critical for accurately modeling IVOC emissions and their impact on aerosol formation. To capture this variability, we distinguish between ten global regions (Fig. 3, Table 2), aligned with the IPCC AR6 regional breakdown (Szopa et al., 2021; Tsimpidi et al., 2025). For each region, we compiled available country-level data on the relative shares of diesel and gasoline consumption during the years 2010–2020. For Latin America and the Caribbean, Africa, and Europe, regional information was derived from 20, 34, and 27 countries, respectively. For the Middle East, Asia-Pacific Developed, and Southeast Asia and Developing Pacific regions, available data from a few countries in each region were averaged to represent the whole region. For other regions (i.e., Eastern Asia, South Asia, Eurasia, and North America), data from a single country (i.e, China, India, Russia, and the USA, respectively) were selected, as these countries already dominate the emissions of their respective regions. The regional average fuel shares are expressed as the fraction of diesel or gasoline fuel consumed per unit of total fuel (diesel + gasoline), denoted as $d_{fuel}$ and $g_{fuel}$, respectively. However, since diesel and gasoline have different VOC emission factors (2054 mg VOC L$^{-1}$ for diesel and 3382 mg VOC L$^{-1}$ for gasoline (Zhao et al., 2015; 2016), the fuel volume shares do not directly translate into VOC emission shares. For example, if a country consumes equal volumes of diesel and gasoline ($d_{fuel} = g_{fuel}$), diesel accounts for only ~38% of the total VOC emissions due to its lower emission factor. The diesel VOC share (D$_{VOC}$) and gasoline VOC share (G$_{VOC}$) are therefore calculated as follows:

$$\text{D}_{\text{VOC}} = \frac{d_{fuel} \cdot 2054}{d_{fuel} \cdot 2054 + g_{fuel} \cdot 3382}, \qquad \text{G}_{\text{VOC}} = 1 - \text{D}_{\text{VOC}} \qquad \text{(Eq. 1)}$$

The resulting gasoline and diesel contributions to the total fuel consumed and to the total road transport-related VOC emissions for each subcontinent are summarized in Table 2. Figure 3 illustrates the regional variation in the diesel share of VOC emissions. Notably, diesel dominates in Southern Asia and Europe, whereas gasoline is the primary fuel in North America. Finally, the calculated $d_{fuel}$ and $g_{fuel}$ ratios are applied to the road transport VOC inventory, allowing to generate two separate VOC inventories, one for diesel and one for gasoline emissions, in units of molecules m$^{-2}$ s$^{-1}$. These inventories are then used to apply fuel-specific IVOC emission factors within the ORACLE-IVOC framework.

**Table 2:** Share of diesel and gasoline in total fuel consumption and total VOC emissions from the road transport sector.

| Continent | Diesel (%) ($d_{fuel}$) | Gasoline (%) ($g_{fuel}$) | Years | Source | Diesel VOC (%) ($D_{VOC}$) | Gasoline VOC (%) ($G_{VOC}$) |
|---|---|---|---|---|---|---|
| 1. Southern Asia | 78 | 22 | 2010 – 2019 | Bhatt, 2021 | 68 | 32 |
| 2. Europe | 73 | 27 | 2010 – 2020 | Fuels Europe, 2022 | 62 | 38 |
| 3. Eastern Asia | 64 | 36 | 2010 – 2016 | He et al., 2017; Li et al., 2020 | 52 | 48 |
| 4. Africa | 57 | 43 | 2010 – 2016 | Liddle and Huntington, 2020 | 45 | 55 |
| 5. Middle East | 52 | 48 | 2010 – 2016 | Ghorbani et al., 2018; Rahman et al., 2022 | 40 | 60 |
| 6. Asia-Pacific Developed | 49 | 51 | 2010 – 2020 | Australian Bureau of Statistics, 2020; Klein, 2025a, b | 37 | 63 |
| 7. Latin America & Caribbean | 46 | 54 | 2010 – 2016 | Liddle and Huntington, 2020 | 34 | 66 |
| 8. SE Asia & Developing Pacific | 40 | 61 | 2013 – 2018 | Xie and Harjono, 2020; Liu and Lin, 2019 | 28 | 72 |
| 9. Eurasia | 38 | 62 | 2015 | Grushevenko et al., 2018 | 27 | 73 |
| 10. North America | 25 | 75 | 2010 – 2020 | Statista Research Department, 2025 | 17 | 83 |

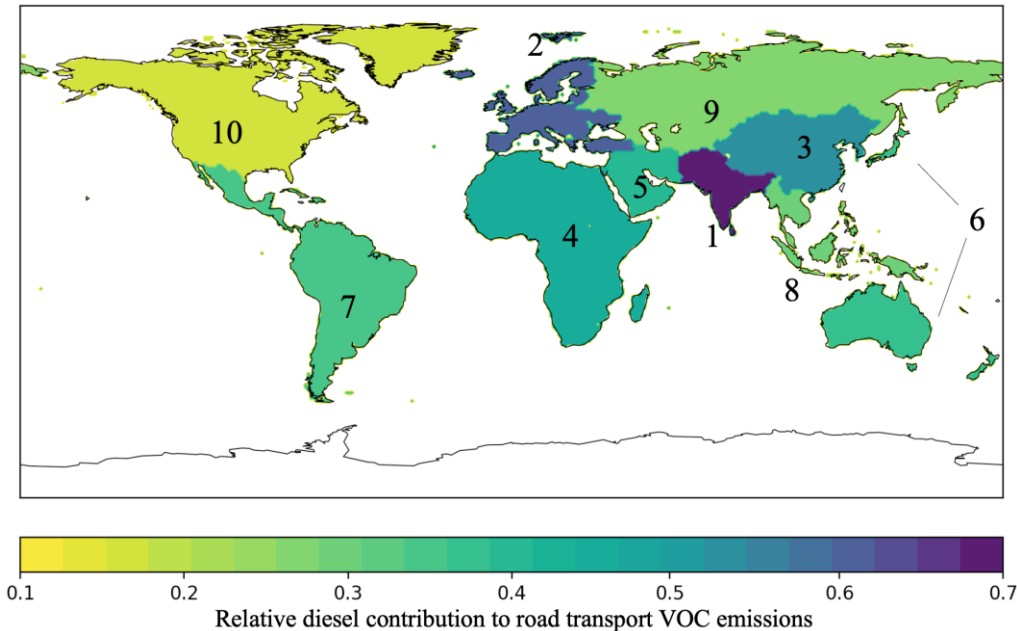

**Figure 3:** Global diesel share of VOC emissions in 10 subcontinents. The numbers correspond to the subcontinent names in Table 2.

### 2.3.2 IVOC emission factor dataset

The IVOC emission factors (EF) for diesel combustion were taken from Zhao et al. (2015), who conducted controlled experiments on three heavy duty vehicles (HDDV) and two medium duty vehicles (MDDV) from model years 2001 to 2010. Tests were conducted using four different diesel fuels and a range of driving cycles, including hot start and cold start urban driving, as well as low and high speed HDDV operation. For gasoline, the IVOC emission factors were taken from Zhao et al. (2016), who tested 42 light duty gasoline vehicles (LDGV) spanning model years 1984–2012. The fleet included diverse vehicle types, engines, and aftertreatment technologies. Emission data covered cold-start and hot start united cycles, as well as high-speed freeway cycles. This broad set of test conditions is crucial, as both IVOC composition and total emissions are influenced by temperature, driving cycle, and vehicle technology (Zhao et al., 2015; Zhang et al., 2020; Hartikainen et al., 2023; Paul et al., 2024). Further methodological details are provided in May et al. (2014), and Gordon et al. (2014).

The tested vehicles represent mostly US vehicle standards built before 2012, whereas ORACLE-IVOC aims to simulate global emissions for the entire 2010s decade. Consequently, these emission factors can only serve as a preliminary approximation. For instance, IVOC/VOC emission ratios from gasoline combustion tend to increase with stricter vehicle emission standards (Liu et al., 2021; Drozd et al., 2018), suggesting that while overall emissions may decrease, the proportion of IVOCs within VOCs may rise. Fleet compositions and emissions regulations also differ substantially across continents. For example, according to a 2017 report by the Chinese Ministry of Ecology and Environment, only 5% of diesel cars were equipped with an aftertreatment system (Liu et al., 2021). On the other hand, Sarica et al. (2023), using a methodology that accounted for the full range of pre-Euro to Euro-6 vehicles, found an average IVOC/VOC ratio of 31% for diesel combustion. This ratio is much lower than the average ratio of 90% found by Zhao et al. (2015) due to uncertainties in measurements with diesel particle filters. For gasoline vehicles, the IVOC/VOC ratios used in ORACLE-IVOC (4% and 17% for cold and hot starts, respectively; Zhao et al., 2016) align well with those reported by Sarica et al. (2023) who found a 6% ratio for cold start cycles across Euro standards. Similarly, Zhang et al. (2022b) tested vehicles conforming to China 6 standards (comparable to Euro 6) and found IVOC/VOC ratios of 4% and 25% for cold and hot starts, respectively. Despite the variability in IVOC/VOC ratio, the chemical composition of IVOCs appears to be relatively consistent across different vehicles and test conditions for the same fuel type (Zhao et al., 2015; Drozd et al., 2018; Lu et al., 2018). Therefore, applying the IVOC composition from Zhao et al. (2015; 2016) globally for the 2010s is a reasonable first-order approximation, despite uncertainties in emission ratios, particularly for diesel. These studies provide detailed data on both speciated IVOCs and unspeciated compounds, categorized by carbon number and aromatic or aliphatic structure. This chemical detail enables consistent mapping to the lumped species framework of ORACLE-IVOC across all regions and simulation years, using existing global VOC emission inventories. Such consistency is essential for comparing the new approach to the previous OC-based method in ORACLE-base and evaluating its impact on global SOA formation. The impact of emission uncertainties, associated with the variability in IVOC/VOC ratios due to differences in vehicle types, emission standards, and driving conditions, on modeled SOA formation will be analyzed in a future study.

**3 The IVOC emission scenario in ORACLE-IVOC for the road transport sector**

**3.1 Global distribution of IVOC emissions**

The IVOC emissions from diesel and gasoline vehicles calculated based on the VOC inventory by utilizing the emission factors of Zhao et al. (2015; 2016) are shown in Fig. 4. As expected, the emission peaks are in metropolitan areas around the globe, with the highest emissions calculated in Eastern China. A large and dense cluster of high emissions is also found in Iraq and Western Iran. Given the global demand for transportation across most inhabited areas, organic compound emissions from road transport are widely distributed, except for the remote regions of Alaska, Siberia, the Amazon basin, the Sahara and Sahel zones, and the Australian outback. In these regions, emissions are largely limited to major highways, which appear distinctly on the emission maps. Even in more densely populated countries, the major roads and cities clearly show higher emissions than the surrounding areas. This spatial contrast is particularly striking for diesel emissions along the ring highway in Afghanistan (Fig. 4a). The Gulf region of Iran, Turkmenistan, and Pakistan also stand out in this regard for both fuel types. In contrast, the distribution of emissions in North America is characterized by a dense and more uniform pattern throughout the entire continent. Due to the heavily gasoline dominated fleet distribution (accounting for 83% of road transport-VOC emissions), the US also has the most pronounced contrast between diesel and gasoline emissions of any region. The second most noticeable difference in the distribution is the emission-rich region of Southern Asia, which has the highest share of diesel consumption (accounting for 68% of road transport-VOC emissions) among the 10 regions considered.

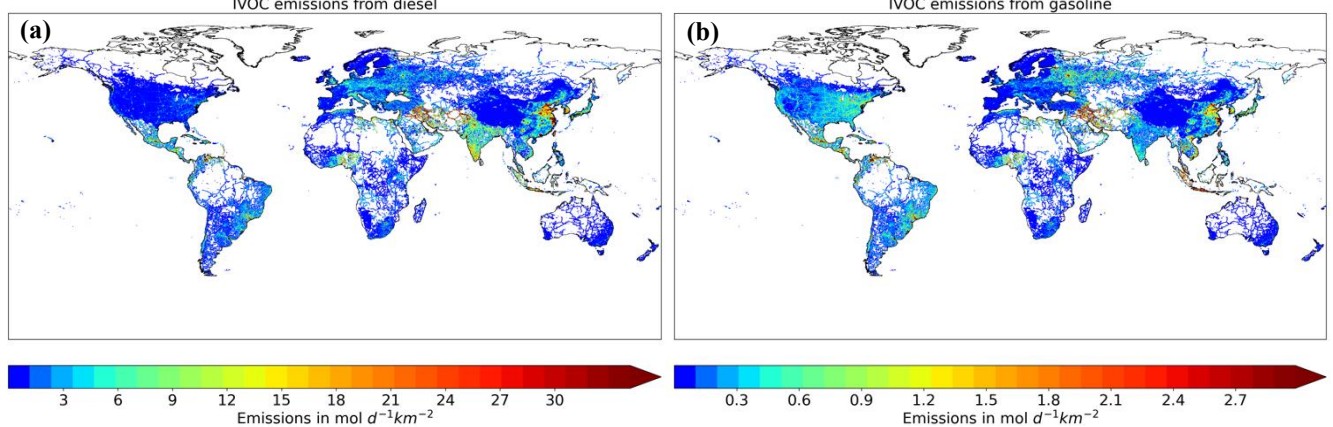

**Figure 4:** IVOC emissions from the road transport sector used in ORACLE-IVOC, averaged over the years 2010 – 2020. (a) Emissions from diesel engine vehicles and (b) emissions from gasoline engine vehicles, both derived proportionally to VOC emissions.

The amount of the emitted IVOCs from diesel and gasoline combustion differs dramatically. Diesel IVOC emissions are about 1 order of magnitude higher than gasoline emissions, even in countries where the gasoline consumption is much higher than the diesel consumption, such as in North America, Eastern Eurasia, and the Developing Pacific region (Table 2, Fig. 3).

This can be attributed to the significantly higher IVOC/VOC molar ratio for diesel emissions (0.31) compared to gasoline (0.02).

Comparing the updated total IVOC emissions (i.e., from diesel and gasoline) derived from the VOC inventory using the emission factors of Zhao et al. (2015, 2016) (hereafter referred to as $IVOC_{VOC}$) with the previous approach that estimated IVOC emissions as 1.5 times the primary OC inventory (hereafter referred to as $IVOC_{1.5OC}$), reveals stark increases across many regions (Fig. S2). In Nigeria, Ukraine, Russia, and the Caribbean, $IVOC_{VOC}$ emissions are up to 200 times higher than $IVOC_{1.5OC}$; in the U.S., about 50 times higher; and in Western Europe, only 2–10 times higher. This variation reflects global differences in diesel versus gasoline consumption. Diesel combustion produces more particulates and less volatile organic compounds (i.e., more OC), while gasoline combustion emits more high-volatility gaseous compounds (i.e., more VOCs) and negligible OC (Lu et al., 2018; Gentner et al., 2017; Morino et al., 2022). As a result, estimating gasoline-related IVOC emissions based on scaled OC significantly underrepresents them. Furthermore, recent studies (Zhao et al., 2015; 2016; Jathar et al., 2017; Lu et al., 2018) have shown that IVOC emissions correlate more strongly with VOCs than with OC. Consequently, countries with high gasoline consumption show much larger increases under the VOC-based method. Although total IVOC emissions from diesel are higher (Fig. 4a), the relative increase in gasoline-related emissions is more pronounced (Fig. S2). In Europe, where diesel dominates, higher OC and lower VOC emissions result in a comparatively smaller increase, consistent with the origins of the $IVOC_{1.5OC}$ method, which was based on diesel exhaust measurements (Robinson et al., 2007). Thus, regions with high diesel use show smaller discrepancies between the two approaches, while gasoline-dominated regions show much larger deviations, underscoring the need for fuel-specific IVOC emission modeling.

## 3.2 Chemical composition of IVOC emissions

In addition to the difference in magnitude of IVOC emissions from gasoline versus diesel combustion, the difference in the IVOC composition of gasoline versus diesel exhaust is also noteworthy. Figure 5 shows the lumped IVOC composition for diesel and gasoline emissions, based on the sum of the emission factors assigned to each of the lumped species. The total emissions for Europe, North America, Eastern Asia and Southern Asia are shown for the entire area of each region. Since the gasoline and diesel emission factors are assumed not to vary between the continents, the composition discussed here is the same for all four continents.

The diesel IVOC emission spectrum is dominated by alkanes which account for approximately 98% of total emissions. Within this group, the lumped species ALK6 contributes about 50%. PAH1 make about 1.5% of diesel IVOCs, while heavier PAH2 and aromatic species are almost negligible. In contrast, the IVOC composition from gasoline exhaust is more diverse with significantly more aromatics. The more volatile PAH1 species dominate the gasoline IVOC spectrum, accounting for 67% of emissions, mainly due to the high emission factors for naphthalene and other unspeciated PAHs. The ALK6 emissions are also important with a share of 16% to the IVOC composition. The other alkane lumped species together account for an additional 11% but individually contribute less than PAH2 (6%). The lumped aromatics ARO3 play an insignificant role in

the IVOC emission spectrum of both diesel and gasoline with 0.1 and 0.2%, respectively. However, they should not be neglected for SOA formation due to their high SOA yields (Table 1).

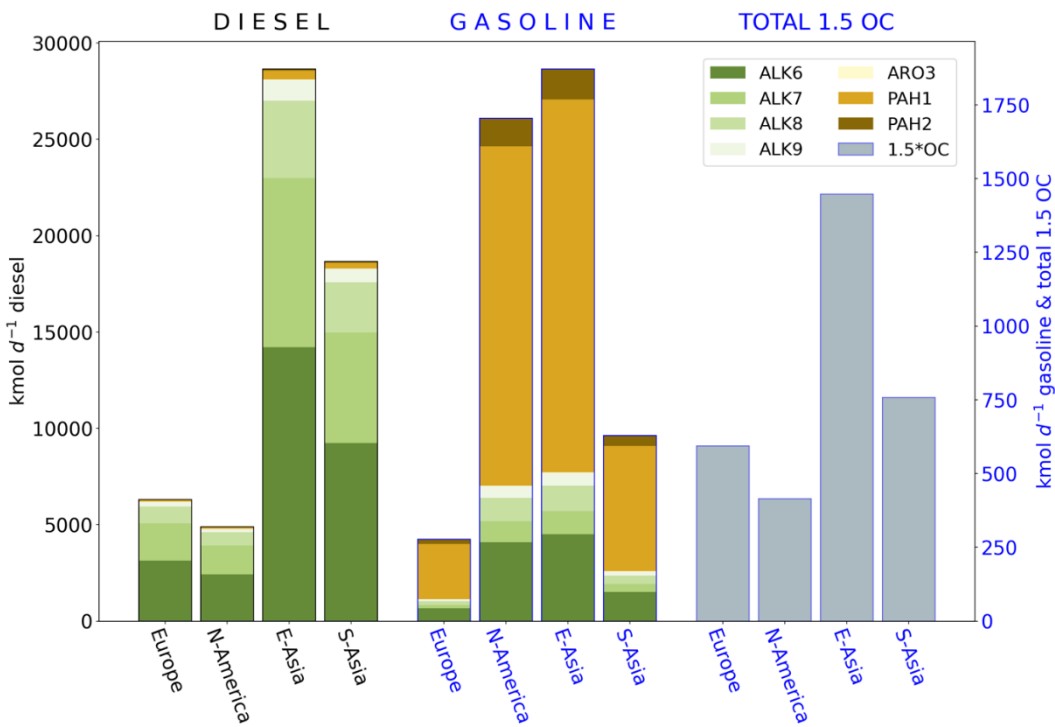

**Figure 5:** Average IVOC emission fluxes (in kmol d[-1]) from the road transport sector in Europe, North America, Eastern and Southern Asia over the 2010–2020 period in the ORACLE-IVOC model. The contributions of the 7 lumped species for diesel (left) and gasoline (middle) using the new VOC-based approach in ORACLE-IVOC, and the IVOC emissions derived from OC emissions using the VBS approach used ORACLE-base (right) are shown. Different y-axis scales are used for the three emission estimates.

For comparison, Fig. 5 also shows calculated IVOC emissions derived from a factor of 1.5 applied to OC emissions ($IVOC_{1.5OC}$) used in the previous ORACLE model. In this approach, gasoline and diesel emissions are not distinguished, and only the combined emission load is presented. The $IVOC_{1.5OC}$ emissions are about 1 order of magnitude lower than the diesel IVOC emissions derived from VOCs ($IVOC_{VOC}$) alone: Eastern Asia exhibits the highest daily emissions, with about 1,500 kmol d[-1] for $IVOC_{1.5OC}$ compared to 28,700 kmol d[-1] for diesel $IVOC_{VOC}$. North America has the lowest daily average emissions, with about 400 kmol d[-1] $IVOC_{1.5OC}$ versus 4,900 kmol d[-1] diesel $IVOC_{VOC}$. However, the relative proportions between the four continents of the diesel emission loads and the $IVOC_{1.5OC}$ loads are very similar. This is most likely due to the close relationship between the here chosen multiplier of 1.5 times the OC emissions and diesel emissions in general, as explained in Section 3.1.

Comparing $IVOC_{VOC}$ from gasoline combustion with $IVOC_{1.5OC}$, the overall emission levels are similar in magnitude, but the regional distributions differ notably. Eastern Asia leads in both cases, but gasoline $IVOC_{VOC}$ emissions (1,873 kmol d$^{-1}$) are almost 30% higher than those of $IVOC_{1.5OC}$. These relatively high gasoline $IVOC_{VOC}$ emissions are observed as emission hotspots across Eastern China and South Korea (Fig. 4b). North America follows closely with gasoline $IVOC_{VOC}$ emissions of 1704 kmol d$^{-1}$, owing to its high gasoline share in VOC emissions (83%). Therefore, we also find that North America has the smallest difference between the amount of gasoline and diesel IVOC emissions. Here, diesel $IVOC_{VOC}$ emissions are only three times higher than gasoline $IVOC_{VOC}$ emissions, a much smaller gap than in other regions. In contrast, Southern Asia, despite being one of the most populous and polluted regions globally, shows substantially lower gasoline $IVOC_{VOC}$ emissions (600 kmol day$^{-1}$), which is about one-third of the values in North America and Eastern Asia. This discrepancy reflects the low gasoline contribution to VOC emissions in South Asia (i.e., only 32%), which is the lowest among the four regions. In contrast, the diesel $IVOC_{VOC}$ emissions in South Asia are consequently very high – with 18,700 kmol d$^{-1}$ 31 times higher than gasoline $IVOC_{VOC}$ emissions and the second highest amount of emissions of the presented four continents. Europe, with a similarly low gasoline VOC share of 38%, has the lowest gasoline $IVOC_{VOC}$ emissions among the four regions (300 kmol d$^{-1}$). This is 23 times lower than Europe's diesel $IVOC_{VOC}$ emissions of 6,300 kmol d$^{-1}$.

**4 Results and discussion: Simulated OA from road transport**

Using both the newly developed ORACLE-IVOC and the previously employed ORACLE-base model, a 10-year simulation (2010–2020) was conducted that considers the chemical evolution of all organic compounds, including LVOC, SVOC, IVOC, and VOCs of both biogenic and anthropogenic origin. To evaluate the impact of the new modeling framework, we used the global dataset compiled by Tsimpidi et al. (2025), which includes AMS measurements from field campaigns conducted worldwide during the 2010s. Simulated total SOA was compared against PMF-derived oxygenated organic aerosol (OOA) from this dataset. Although traffic emissions contribute only partially to total SOA, the implementation of ORACLE-IVOC led to a modest but consistent improvement in model performance (Table S1, Fig. S3). The average underprediction of observed SOA concentrations decreased from 63% (ORACLE-base) to 61% (ORACLE-IVOC), corresponding to a 6% increase in simulated SOA. While this improvement is relatively small, it is promising, especially given that traffic is not the dominant global IVOC source. Extending the lumped IVOC approach to other sectors with larger IVOC contributions, such as volatile chemical products and biomass burning (Huang et al., 2023), could further enhance model performance. Additionally, we compared ORACLE-IVOC results with CMAQ simulations using the regional IVOC inventory developed by Chang et al. (2022). Unlike ORACLE-base, SOA-iv concentrations simulated with ORACLE-IVOC over Eastern Asia, particularly in regions with medium to high SOA-iv load, were of the same magnitude as CMAQ's OA-iv concentrations attributed to road transport. This agreement adds confidence to the representativeness of our approach, even without full regional differentiation of emissions.

In the following subsections, we focus specifically on OA formation from the three lower volatility groups: IVOCs, SVOCs and LVOCs emitted by the road transport sector. POA is effectively the sum of SVOC and LVOC that partition directly into

the aerosol phase upon emission, without undergoing chemical aging. The SOA is formed through the oxidation of SVOCs (termed SOA-sv) and IVOCs (termed SOA-iv). Thus, *total OA* in this section is defined as POA + SOA-iv + SOA-sv.

## 4.1 SOA-iv from the road transport sector

Figure 6 compares the global 10-year average concentration of SOA-iv from the road transport sector, as calculated by ORACLE-IVOC using a lumping approach and IVOC emissions derived from VOCs (IVOC$_{VOC}$), and the ORACLE-BASE model, which applies a simplified VBS approach using IVOC emissions derived from OC (IVOC$_{1.5OC}$). With ORACLE-IVOC, these concentrations have increased globally by around 1 order of magnitude. Specifically, ORACLE-base yields a global decadal average SOA-iv concentration of 0.007 µg m$^{-3}$ and an atmospheric burden of 0.014 Tg. In contrast, ORACLE-IVOC estimates these values at 0.06 µg m$^{-3}$ and 0.126 Tg, respectively, i.e., about 8 to 9 times higher. This increase is mostly evident in the Northern Hemisphere, where we have an increase factor of about 10, while the SOA-iv increase over the Southern Hemisphere oceans is only half as large. The reason for this is that most of the emission sources are in the Northern Hemisphere, but it is rare for emissions to be transported across the equator because the mixing between the two hemispheres is slow (Kling at Ackerly, 2020).

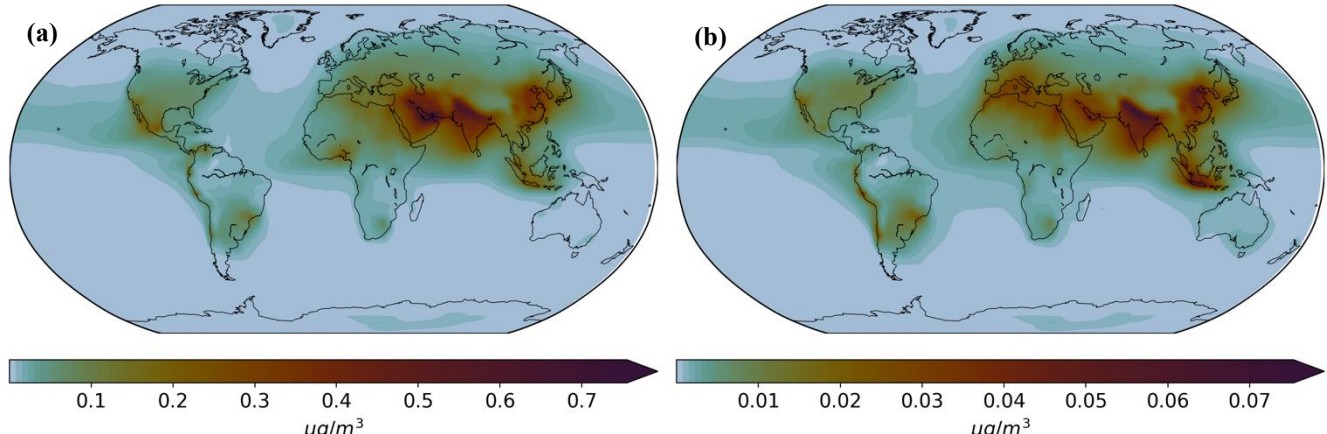

**Figure 6:** Modelled surface concentration of SOA-iv from the road transport sector for the years 2011 – 2020 by using (a) the ORACLE-IVOC model and (b) the ORACLE-base model.

The highest SOA-iv concentrations simulated with ORACLE-IVOC are over the Middle East, Southern and Eastern Asia (Fig. 6a). In the ORACLE-base simulation, while Southern and Eastern Asia also exhibit high concentrations, the Middle East show slightly lower levels and the Developing Pacific islands show relatively higher concentrations (Fig. 6b). The maximum SOA-iv concentration by ORACLE-base is 0.075 µg m$^{-3}$, while the ORACLE-IVOC maximum is 0.76 µg m$^{-3}$. These elevated concentrations of SOA-iv correspond to areas with dense populations and strong road transport sector emissions (NASA, 2018). In addition, the atmospheric and local conditions in these regions are favourable for SOA formation. Over the Himalayas region, including the highly populated regions of Northern India and Bangladesh, where both models predict the highest concentrations, anthropogenic aerosol level are often found to be very high (Tsimpidi et al., 2016; Hassan et al., 2023).

The largest relative increases of SOA-iv concentrations in ORACLE-IVOC occur in areas that are not always the strongest SOA-iv hotspots under ORACLE-base. For example, The SOA-iv values over Nigeria increase by a factor of 37, and by 20–25 times over Mexico and the Caribbean. Similar increases in SOA-iv (~15–20 times higher) occur across parts of the Middle East and Eurasia, corresponding to sharp increases in $IVOC_{VOC}$ emissions compared to $IVOC_{1.5OC}$ (Fig. S4).

In contrast, the lowest increase in a populated area is found over Western Europe, where the concentrations of SOA-iv only doubled with ORACLE-IVOC compared to ORACLE-base. The continental surface burden over all of Europe increased by a factor of 5. Only in remote areas such as the Southern Pacific Ocean are there spots where the SOA-iv increase is just as low (Fig. S4). Consequently, ORACLE-IVOC estimates for Western Europe (~0.05 $\mu g\ m^{-3}$), are rather insignificant in comparison to other higher populated areas (Fig. 6a). Opposed to this, Europe is according to ORACLE-base indeed an important contributor of global SOA-iv, with concentrations ranging from 0.01 to 0.04 $\mu g\ m^{-3}$ (Fig. 6b).

The wide variability in SOA-iv concentrations calculated by ORACLE-IVOC and ORACLE-base is largely driven by the underlying emission assumptions, especially the regional dominance of gasoline versus diesel use. In Nigeria, Mexico, the Caribbean, Eurasia, and also North America, gasoline is the predominant fuel consumed (Table 2, Maduekwe et al., 2020). Accordingly, the ratio of $IVOC_{VOC}$ to $IVOC_{1.5OC}$ is high, as explained in section 3 (Fig. S2). As a result, ORACLE-base underestimates SOA-iv due to a lack of precursor material, while ORACLE-IVOC captures the higher emissions more accurately, leading to much higher concentrations of SOA-iv. Compared to the densely emitting regions of Nigeria, Mexico, and the Caribbean, the more modest increase in SOA-iv concentrations over North America and Eurasia, despite high gasoline consumption share (Table 2, Fig. 3), is likely due to less spatially concentrated emissions (Fig. 4). In addition, tropical regions benefit from enhanced photochemistry, which promotes more efficient SOA formation compared to the northern temperate zone.

In contrast, Europe has the second largest share of diesel vehicles in the world (Table 2, Fig. 3). Consequently, the $IVOC_{1.5OC}$ emissions and the ORACLE-base simulated SOA-iv values are already substantial, and ORACLE-IVOC estimates are not drastically higher (Figs. 4, 5, 6). Even so, due to regional differences (e.g., lower diesel shares in Eastern Europe), the continental surface burden of SOA-iv in Europe still increases by 430% (+ 323 kg) in ORACLE-IVOC. In Southern and Eastern Asia, where diesel consumption is also high, the SOA-iv continental surface burden increases by a factor ~10, which aligns with the global average (Fig. S4). One contributing factor is that in the latter two regions not only are OC emissions exceptionally high, but VOC levels are also much higher compared to Europe (Fig. S1). This provides an abundant source of precursor material for the lumping framework, leading to higher SOA-iv loads. Additionally, the prevailing atmospheric and chemical conditions as enhanced ozone and OH formation in these regions may further enhance SOA-iv formation under the new approach (Chakraborty et al., 2015; Chameides et al. 1992).

Another important observation is the wider spatial distribution of SOA-iv simulated by ORACLE-IVOC compared to ORACLE-base results. In regions of intense emissions and SOA formation (e.g., Asia, West Africa, western North America), ORACLE-IVOC simulates a more widespread aerosol distribution, including enhanced concentrations over adjacent oceanic regions (Fig. 6, Fig. S4). These outflow patterns align with prevailing wind patterns (Kling and Ackerly, 2020). Unlike heavier

aerosols, gaseous IVOCs can be transported more efficiently before oxidizing to SOA-iv., increasing the total OA load far from the original sources. In ORACLE-IVOC, this is facilitated by the lower Henry's law constants used for $IVOC_{VOC}$ species (Table 1), which reflect their more hydrophobic nature similar to the also gaseous VOCs (Jiminez et al., 2009; Pye and Seinfeld, 2010; Hodzic et al., 2014). In contrast, ORACLE-base assumes a much higher $H$ value ($10^5$ M atm$^{-1}$) for $IVOC_{1.5OC}$, similar to the lower volatile SVOCs, and LVOCs, implying rapid wet deposition and limited transport. As a result, $IVOC_{VOC}$ species can travel farther before being removed from the atmosphere or converted to SOA. In addition, the reaction rate constants for IVOCs in ORACLE-IVOC (Table 1), especially those from the alkane-dominated diesel emissions, are lower (e.g., $9.6 \times 10^{-12}$ cm$^3$ molecule$^{-1}$ s$^{-1}$) than those used in ORACLE-base ($2.0 \times 10^{-11}$ cm$^3$ molecule$^{-1}$ s$^{-1}$), meaning that IVOCs have more time to be transported before oxidizing. Due to the different composition of gasoline combustion IVOCs (dominated by aromatics), the reaction rates are on average more similar to the rate used in ORACLE-base. However, their overall contribution is small compared to diesel-derived IVOCs (see Fig. 4), and thus the transport-limiting effect is minimal on a global scale.

This drastic change of SOA-iv concentration and atmospheric load is also evident in both the decadal trend and interannual variability of the atmospheric load (Fig. 7). Despite this difference in magnitude, the seasonality of the SOA-iv load remains nearly identical in both models, with monthly standard deviations of 3–7% in both cases. Concentrations are lowest in winter and higher in spring and autumn when increased sunlight enhances photochemical activity. Both models also predict a slight dip in concentrations during summer, likely due to reduced condensation into the particulate phase at higher temperatures, which counteracts the increase in photochemical activity (Tsimpidi et al., 2016).

Over the decade, the atmospheric burden of SOA-iv increases by 10% in ORACLE-base, but it decreases by 8% in ORACLE-IVOC. Secondary aerosol concentration trends are strongly influenced by changes in the precursor emissions. Data from the CAMS-GLOB-ANT inventory (Soulie et al., 2023) show that emissions from the road transport sector have been declining since 2012, with VOC emissions decreasing by 17% and OC emissions decreasing by only 2.5% by 2020. Given that the atmospheric oxidation capacity has increased due to reductions in some anthropogenic emissions (Tsimpidi et al., 2025), SOA-iv does not decrease as much as its precursors. Therefore, ORACLE-IVOC SOA-iv loads decline less than the precursor $IVOC_{VOC,}$ whereas ORACLE-base SOA-iv can even continue to rise. Despite these different trends, SOA-iv in ORACLE-IVOC remains 8.6 times higher at its lowest point than the highest level recorded in ORACLE-base in 2020.

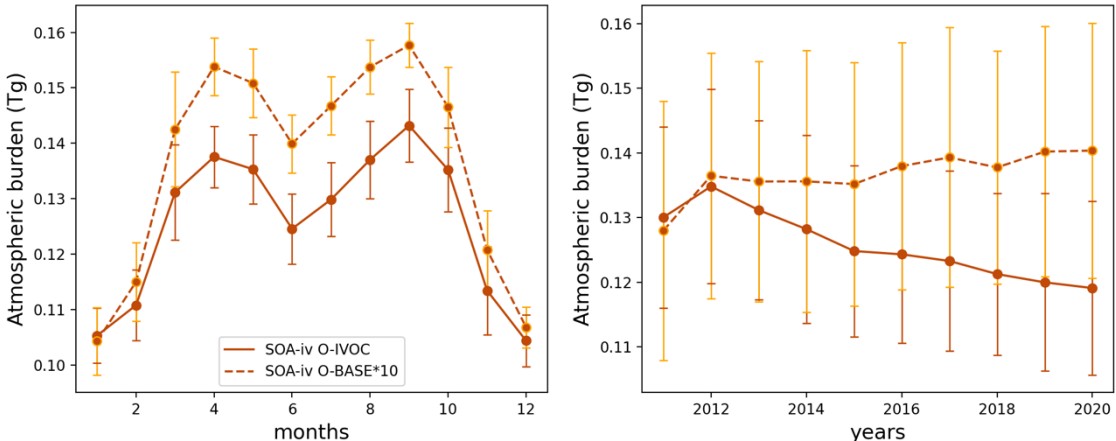

**Figure 7:** Average predicted atmospheric burden of SOA-iv with ORACLE-base and ORACLE-IVOC in comparison. (a) shows the annual variability with monthly averaged loads for the whole decade and (b) shows the decadal variability with annually averaged loads. The error bars are describing the standard variations.


### 4.2 Impact of IVOC emissions on the total OA formation from road transport

Figure 8 shows the global 10-year average surface concentration for *total OA*. Additionally, the *total OA* composition for the 10 different regions is shown in pie charts with the continental surface burden of each POA, SOA-sv and SOA-iv. Overall,
the different approaches in the IVOCs treatment and emission scenarios lead to a nearly 2-fold increase in the maximum OA concentration (0.49 vs. 0.87 µg m$^{-3}$), an almost 4-fold increase in the upper 95$^{th}$ percentile value (0.04 vs. 0.15 µg m$^{-3}$), and a rise in the atmospheric burden from 0.02 Tg in ORACLE-base to 0.13 Tg in ORACLE-IVOC over the decade. Since the evolution of IVOCs, SVOCs, and LVOCs is interconnected, minor differences in absolute POA and SOA-sv concentrations between ORACLE-IVOC and ORACLE-base are expected and observed. Since in ORACLE-IVOC all IVOC$_{VOC}$ are in the
gas phase, POA consists only of particulate SVOCs and LVOCs. In the case of ORACLE-base, the IVOC$_{1.5OC}$ emissions are also initialized as gas-phase emissions, but like SVOCs, they can partition immediately into the aerosol phase and become POA. However, since unoxidized IVOCs exist almost exclusively in the gas-phase (Lu et al., 2018), they contribute only a negligible percentage of $\leq$ 0.1% to POA on most continents. A minor exception is Eastern Asia with a contribution of IVOC in the form of POA to *total OA* of 1.2%. Consequently, the largest absolute change in POA occurs over Eastern Asia, where
ORACLE-IVOC predicts a continental surface burden 5 kg lower than ORACLE-base. However, this corresponds to only a 1.5% decrease. In other regions, differences between the two simulations are 1 kg or less for both POA and SOA-sv, with relative changes mostly below 2.5%.

ORACLE-base predicts the highest *total OA* concentrations over urban regions of China, reaching a maximum of 0.49 µg m$^{-3}$ due to substantial contributions from POA and SOA-sv. According to the ORACLE-IVOC simulation, the *total OA*

concentrations hotspots are with values of up to 0.87 µg m⁻³ located in both Eastern China and again in the northern region of South Asia including the Himalayas, closely followed by the high *total OA* concentrations in the Gulf region. According to ORACLE-base, other regions with more widespread medium-high *total OA* concentrations of 0.15 to 0.3 µg m⁻³ are the Middle East and India, but also Europe, South-East Asia and the Developing Pacific.

Since only the representation of IVOCs in ORACLE-IVOC has changed from ORACLE-base, while the chemistry of
SVOCs and LVOCs remains the same, the changes in the OA composition can be entirely attributed to this modification. Notably, ORACLE-IVOC shows a clear shift towards SOA-iv dominance, with the SOA-iv contribution to *total OA* across continents increasing from 27%–52% in ORACLE-base to 72%–93% in ORACLE-IVOC. Among all 10 continents, Europe has the lowest SOA-iv share in ORACLE-IVOC (72%) and the second lowest in ORACLE-base (32%), with only Eastern Asia showing a lower value. This can largely be attributed to the diesel dominance in European fleet composition, which is
responsible for high OC emissions leading to POA and SOA-sv concentrations above the global average in both ORACLE-IVOC and ORACLE-base.

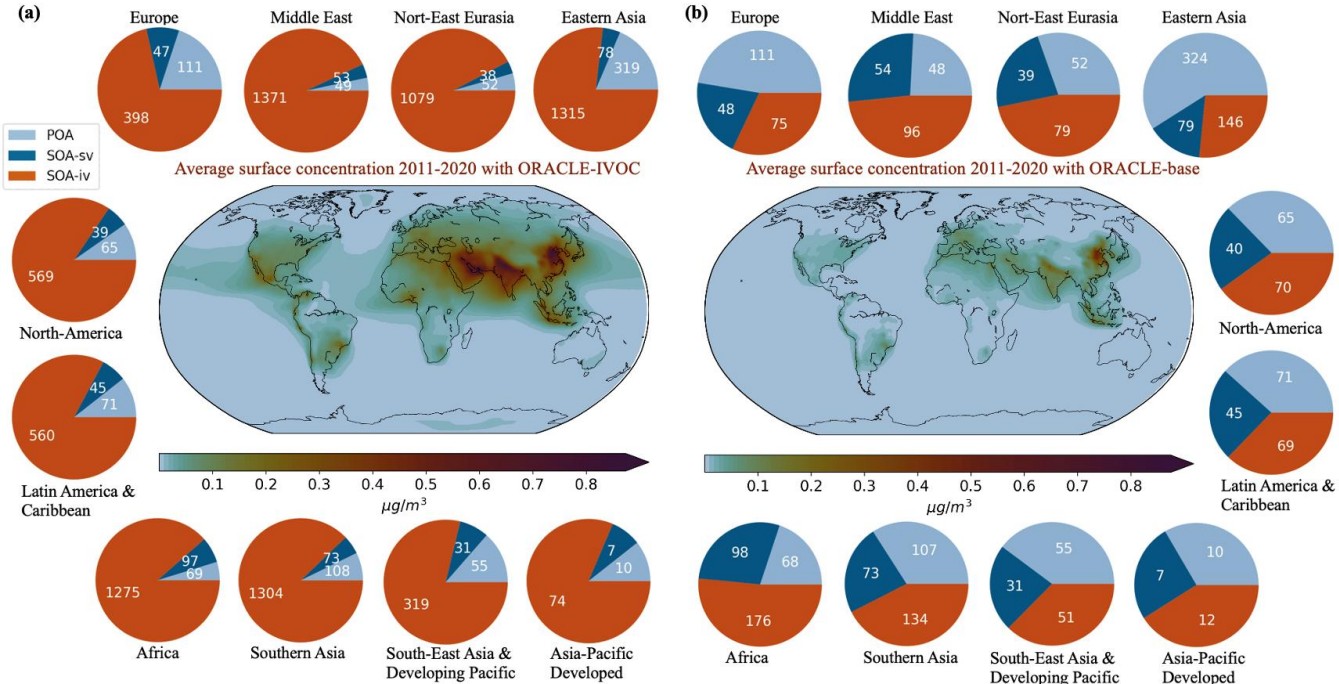

**Figure 8:** *Total OA* surface concentration in µg m⁻³ from road transport (POA + SOA-iv + SOA-sv) as simulated by (a) ORACLE-IVOC and (b) ORACLE-base. The pie charts show the OA composition and continental surface burden in kg in each continent.


Previous studies indicate that SOA-iv remains the dominant contributor to OA formation from diesel combustion. According to Zhao et al. (2015) and Lu at al. (2018), IVOCs can contribute up to 95% and 89% of the total diesel SOA load, respectively. The reason for this high contribution is that diesel combustion exhaust is dominated by a wide range of IVOCs,

most of which are $C_{12} - C_{18}$ species, followed by VOCs and comparatively few SVOCs. In contrast, gasoline is a lighter fuel,
dominated instead by VOC species, and has a narrow range of IVOCs, mostly $C_{12}$ species, which is also well reflected in the
bar plots of Fig. 5 (Gentner et al., 2012; Lu et al., 2018). Therefore, gasoline SOA is dominated by SOA of VOC origin (SOA-
v), and only secondarily by SOA-iv. For example, a study on a modern EURO 6 compliant gasoline car equipped with a
particle filter found that at least 50% of SOA formation could be attributed to VOCs such as toluene, xylene, and
trimethylbenzene alone (Paul et al., 2024). However, since the global fleet primarily consists of older vehicles, the study by
Zhao et al. (2016), which suggests that gasoline-derived SOA can still contain up to 50% SOA-iv, may be more representative.
for global simulations.

Further insights into the differences between the new and previous approaches to simulating SOA-iv can be gained by
comparing the fractional contribution of SOA-iv to *total OA* across both models. Figure 9 shows the fractional difference
between these two ratios, highlighting the spatial variation in how the new lumping-based IVOC treatment alters the SOA-iv
contribution.

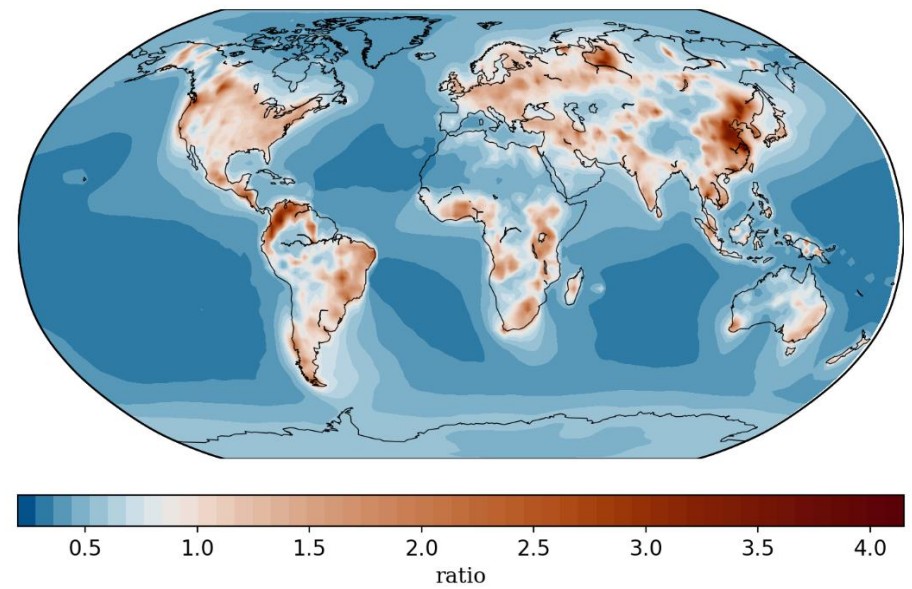

**Figure 9:** Fractional difference in SOA-iv to *total OA* ratio after using the ORACLE-IVOC model.

First, the ratio of SOA-iv in *total OA* from the road transport sector reflects how significantly the new lumping-based IVOC
treatment alters the composition of organic aerosols. While the overall spatial patterns in the SOA-iv–to–OA ratios are broadly
similar between the two models (Fig. S5), the ranges differ markedly. In ORACLE-IVOC, SOA-iv contributes between 35%
and 98% to *total OA*, whereas in ORACLE-base, the contribution ranges from just 10% to 71%. The fractional difference now
between the simulated SOA-iv–to–OA ratios by the two models is uniformly positive across the globe (Fig. 9), indicating that
SOA-iv always contributes more to *total OA* in ORACLE-IVOC than in ORACLE-base. This enhancement is most pronounced

over land, with increases in the SOA-iv contribution reaching up to 400% in some regions. Interestingly, despite these large relative increases, the absolute SOA-iv-to-OA ratios in both models are generally lower over land—particularly near emission sources (Fig. S5). In contrast, ORACLE-IVOC predicts very high SOA-iv fractions (>80%) in many remote and sparsely populated areas, reflecting the longer atmospheric lifetimes and greater transport potential of IVOCs compared to SVOCs and

LVOCs. ORACLE-base also shows somewhat elevated ratios in remote areas (60–70%), but its values are more evenly distributed and consistently lower over land (<40%). The steeper spatial gradient from populated to remote regions in ORACLE-base suggests a more limited transport range for IVOC precursors compared to ORACLE-IVOC, as discussed in Section 4.1. In contrast, the smoother gradients and higher remote-area contributions in ORACLE-IVOC emphasize the improved representation of long-range IVOC transport and oxidation in the updated framework.


### 4.3 Contribution of road transport IVOC to total SOA-iv

To compare SOA-iv from the road transport sector with total SOA-iv, we consider all anthropogenic sectors from the CAMS-GLOB-ANT inventory, including power and other combustion, industry, fugitives, solvents, waste, agriculture, and transport-related sources, as well as biomass burning from the Global Fire Emissions Database (GFED). For all sectors except road

transport, the traditional VBS approach is applied in both ORACLE-IVOC and ORACLE-base, as a complete sectoral differentiation of IVOC emissions would require further development of emission inventories and chemical mechanisms (e.g., Huang et al., 2023), a task beyond the scope of this study but planned for future work. Therefore, the analysis presented here does not provide a full sectoral attribution of SOA-iv, but rather a controlled comparison isolating the impact of the updated road transport IVOC inventory and chemical mechanism.

By incorporating the lumping IVOC approach for road transport, this sector plays a significantly more important role in the global SOA-iv budget in ORACLE-IVOC compared to the traditional ORACLE-base (Fig. 10). In ORACLE-IVOC, the average SOA-iv contribution of the road transport sector to total anthropogenic SOA-iv is 35%, with peak values reaching 98% in the Gulf region, including Iran. Other regions with notably high contributions from road transport to total SOA-iv load include the Caribbean, Mexico, North Africa, Europe, Eastern Asia, and Japan. In contrast, ORACLE-base attributes a much

smaller role to road transport, with an average contribution of only 3%. The greatest influence of road transport to total SOA-iv formation appears over Europe, North Africa, and the Middle East (up to 15%), followed by the west coast of South America (~10%). Despite the large differences in magnitude, both models agree on the geographic regions where road transport emissions are relatively more important for SOA-iv formation.


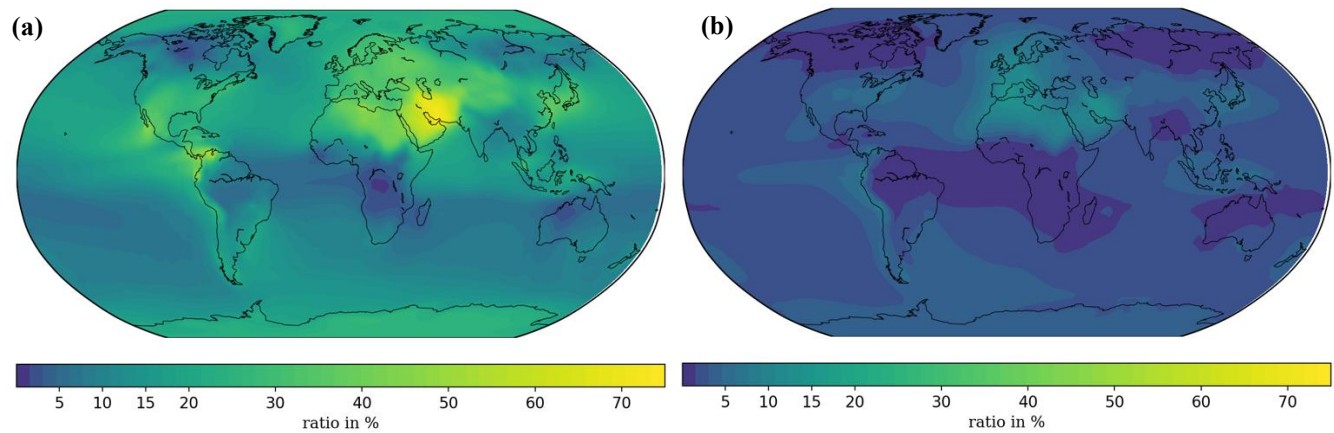

**Figure 10:** Ratio of modelled SOA-iv from the road transport sector to SOA-iv from all sectors as simulated by the (a) ORACLE-IVOC and (b) ORACLE-base, averaged over the years 2011–2020.

Considering that the multiplier of 1.5 of OC emissions is derived from experiments with diesel (Robinson et al., 2007), it is likely that this approximation is even less appropriate for other sectors and may well be higher or lower. Certainly, the neglected complexity of IVOC speciation leads to further underrepresentation of SOA-iv from other sectors. The most relevant sectors in this regard, due to both emission magnitude and IVOC content, include volatile chemical products, domestic combustion, biomass burning, and ship emissions (Huang et al., 2023; Hatch et al., 2017; Huang et al., 2018; Tang et al., 2022; Huang et al., 2022; Kangasniemi et al., 2023). Expanding the lumped IVOC approach to include these and other sectors will likely affect both the magnitude and spatial distribution of the road transport sector's contribution to total SOA-iv, compared to the current results shown in Fig. 10a.

## 5 Conclusion

This study presents the incorporation of a lumped species approach into the organic aerosol module ORACLE to simulate IVOC emissions from the road transport sector and their impact on SOA formation and global OA load. The new approach is novel in three key aspects. First, it accounts for the complexity of IVOC emissions and their variability across different sources by introducing seven lumped species based on 79 IVOC/VOC emission factors for diesel and gasoline internal combustion engines. These factors are derived from chemical structure, molecular weight, carbon number, and reaction rate. Gasoline is dominated by lighter PAH IVOCs with higher reaction rates and lower yields compared to diesel, which is dominated by alkanes with lower reaction rates but higher yields. Second, the emission factors are linked to VOC emissions from the CAMS database that has been modified to account for regional variations in diesel and gasoline consumption across 10 global regions. Third, the reactions of the seven lumped species with OH to form secondary organic gases are implemented, considering

individual reaction rates for alkanes and aromatics and high-NOx SOA mass-based yields for five different saturation concentrations ($10^{-1}$ to $10^3$ µg m$^{-3}$).

These updates have significantly altered both the IVOC emissions from the road transport sector and the simulated SOA formation compared to the ORACLE-base, where IVOC emissions are approximated by scaling the OC emissions. The new IVOC emission factors, combined with the generally higher VOC concentrations than OC, result in an average increase in IVOC emissions by 1 order of magnitude. In regions with high gasoline consumption, IVOC concentrations can be 50 to 200 times higher, even though the IVOC emission factors for gasoline are about 1 order of magnitude lower than for diesel. This is because gasoline exhaust is dominated by light VOCs and almost free of particulates, resulting in low OC concentrations in these areas. In contrast, diesel emissions contain both VOCs and OC, and the scaling factor of 1.5×OC is based on diesel engine measurements. This makes the ORACLE-base approach comparatively more suitable for diesel-dominated regions such as Northern and Western Europe, where IVOC concentrations are only 2 to 10 times higher than those based on IVOC$_{1.5OC}$.

The increase in SOA-iv concentrations in the ORACLE-IVOC simulation is accordingly most pronounced in regions with high gasoline consumption (e.g., North America), where the increase is a factor of 15 to 35. The east coast of Asia shows the highest fractional increase in SOA-iv contributions to *total OA* (~400%), followed by Latin America. These regions also have the highest OA concentrations in the ORACLE-base model, followed by Southern Asia and the Developing Pacific. Moreover, SOA-iv in the Middle East (Gulf region) becomes much more prominent in the ORACLE-IVOC simulation due to the increase in IVOCs associated with gasoline emissions that were significantly underestimated in ORACLE-base. In contrast, the SOA-iv simulated by ORACLE-IVOC in Europe only increased by a factor of 2, the lowest increase of all 10 continents, due to the high diesel share in the region and therefore low VOC emissions. In Southern Asia, the region with the highest diesel shares and the highest SOA-iv concentrations in the road transport sector, the SOA-iv burden still increases by a factor of 10, in line with the global average. This increase is partly because the VOC emissions in Southern Asia are not only higher than in Europe in absolute terms, but also higher in relation to OC. The reasons for this are most likely the differences in emission standards of the fleets in the two continents. Consequently, the very high IVOC$_{VOC}$ emissions can be converted into significant SOA-iv under the new lumping framework. Globally, the atmospheric burden of SOA-iv increased from 0.014 Tg to 0.13 Tg in the ORACLE-IVOC simulation, an almost 10-fold increase. Meanwhile, POA and SOA-sv concentrations remain largely unchanged in both model versions, contributing only marginally to the OA load from road transport. A comparison of SOA-iv from road transport with all other anthropogenic sectors reveals a new dominance of the road transport sector, which contributes 35% to the global SOA-iv burden in ORACLE-IVOC, compared to only 3% in ORACLE-base. However, this result is still influenced by the 1.5×OC scaling method applied to emissions from other sectors, which obscures the true role of road transport relative to other IVOC-rich sectors such as shipping, biomass burning, and certain industries. Nevertheless, given the significant role of diesel fuel in IVOC emissions and the contribution of road transport to global VOC emissions (17% of total anthropogenic VOCs, not including biomass burning), it is likely that road transport will continue to play a crucial role in the global SOA-iv and total OA landscape, even with higher IVOC concentrations and yields from other sectors.


**Code and data availability**. The usage of MESSy (Modular Earth Submodel System) and access to the source code is licensed to all affiliates of institutions which are members of the MESSy Consortium. Institutions can become members of the MESSy Consortium by signing the MESSy Memorandum of Understanding. More information can be found on the MESSy Consortium Website http://www.messy-interface.org. The code developed in this study is archived with a restricted access

DOI (https://doi.org/10.5281/zenodo.15474860) and has already been incorporated into the official development branch of the EMAC modelling system and will therefore be part of all future released versions.

**Authors contribution:** APT designed the research with contributions from VAK and SNP. SS and APT developed the ORACLE-IVOC module. SS selected all information and data needed for the implementation of the lumped IVOC approach

in ORACLE with contributions from APT and SNP. SS performed the simulations and analyzed the results. SS and APT wrote the manuscript with contributions from VAK, HF, and GG. All co-authors made revisions and corrections.

**Competing interests:** The authors declare that no competing interests are present.

**Acknowledgements:** The work described in this paper has received funding from the Initiative and Networking Fund of the Helmholtz Association through the project "Advanced Earth System Modelling Capacity (ESM)". The authors gratefully acknowledge the Earth System Modelling Project (ESM) for funding this work by providing computing time on the ESM partition of the supercomputer JUWELS (Alvarez, 2021) at the Jülich Supercomputing Centre (JSC).

**Financial support:** This research has been supported by the project FORCeS funded by the European Union's Horizon 2020 research and innovation program under grant agreement no. 821205.

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
