# Peer review of "Incorporation of lumped IVOC emissions into the ORACLE model (V1.1): A multi-product framework for assessing global SOA formation from internal combustion engines"

_EGUsphere, 2025_

## Author Comment (AC1)

**Authors' response to comments made by anonymous reviewer #2:**

**Summary**

In this study, the authors present an update to the chemistry module ORACLE by incorporating IVOC emissions from the road transport sector. It turns out the update made significant changes to the simulation OA (e.g., Fig 9). I think this is a high-quality study; the work is well written. I think it fits GMD relatively well. And it is of general interest to the community. I only offer two minor substantive comments below. These are non-blocking and I therefore support the publication of this work. For the record, I found Reviewer 1's comments to be worthy of consideration and I encourage the authors to address them.

We would like to thank the reviewer for his/her review and positive response. Below is a point-by-point response (in black) to all comments raised (in blue).

**Comments**

1. L124: Not sure if the ", but" is needed here? And maybe perspectives (plural) at the end of the sentence?

Thank you. The sentence has been revised accordingly.

2. L138: Can you say more why you chose not to discuss SOA formation? It's not readily clear to me that it is not related. You are not saying ultimate SOA load in both versions will be the same, right? Maybe L323 is related? Planned future work?

We thank the reviewer for this thoughtful question. ORACLE tracks SOA formation from VOC, IVOC, and SVOC precursors separately. While the treatment of IVOC and SVOC emissions and chemistry from road transport differs between ORACLE-base and ORACLE-IVOC, the SOA formation from VOCs is identical in both configurations, following the implementation described by Tsimpidi et al. (2014). As such, VOC-derived SOA does not influence the comparison between the two model versions. Including total SOA in the comparison would mask the differences specifically attributable to the updated IVOC emissions and chemistry, which are the focus of this study. For this reason, we chose not to discuss total SOA formation in detail. We have clarified this rationale in the revised manuscript.

3. L617: Uhh. Didn't you say you didn't want to discuss SOA formation? Am I confused?

While we do not address SOA formation from traditional VOCs in this work, we do evaluate the impact of IVOC emissions on SOA formation (referred to as SOA-iv) and analyze how the newly implemented lumped species approach for IVOCs emitted by the traffic sector influences the calculated SOA-iv concentrations.

4. Code/data: Not sure if this aligns with GMD's policies, but I defer to the editor. As a curious reviewer, I couldn't see the code and I couldn't see underlying data. The underlying data isn't even cited in this statement. Nbd on my end, but just pointing it out to the editor.

Since the usage of MESSy and access to the source code is licensed to all affiliates of institutions which are members of the MESSy Consortium, the code developed in this study is archived with a restricted access DOI https://doi.org/10.5281/zenodo.15474860 (The MESSy Consortium, 2025) and is available by request to the MESSy consortium. The data produced in the study are available from the author upon request. This information has been included in the revised "code and data availability" section.

**References**

The MESSy Consortium: The Modular Earth Submodel System (v2.55.2\_1009-IVOC\_12ff6f14), Zenodo, <a href="https://doi.org/10.5281/zenodo.15474860">https://doi.org/10.5281/zenodo.15474860</a>, 2025

Tsimpidi, A. P., Karydis, V. A., Pozzer, A., Pandis, S. N., and Lelieveld, J.: ORACLE (v1.0): module to simulate the organic aerosol composition and evolution in the atmosphere, Geoscientific Model Development, 7, 3153-3172, 10.5194/gmd-7-3153-2014, 2014.